# Immunological Study of Combined Administration of SARS-CoV-2 DNA Vaccine and Inactivated Vaccine

**DOI:** 10.3390/vaccines10060929

**Published:** 2022-06-10

**Authors:** Ziyan Meng, Danjing Ma, Suqin Duan, Jingjing Zhang, Rong Yue, Xinghang Li, Yang Gao, Xueqi Li, Fengyuan Zeng, Xiangxiong Xu, Guorun Jiang, Yun Liao, Shengtao Fan, Zhenye Niu, Dandan Li, Li Yu, Heng Zhao, Xingli Xu, Lichun Wang, Ying Zhang, Longding Liu, Qihan Li

**Affiliations:** Yunnan Key Laboratory of Vaccine Research and Development on Severe Infectious Diseases, Institute of Medical Biology, Chinese Academy of Medical Sciences & Peking Union Medical College, Kunming 650118, China; mengzy0724@163.com (Z.M.); ynyxmdj@126.com (D.M.); duansuqin@imbcams.com.cn (S.D.); zhangjingjing940115@imbcams.com.cn (J.Z.); 13265173441@163.com (R.Y.); lixinghangsir@163.com (X.L.); gaoyang961107@163.com (Y.G.); angelkiki@imbcams.com.cn (X.L.); zengfengyuan0120@163.com (F.Z.); xiangxiongxu@mail.ynu.edu.cn (X.X.); jgr@imbcams.com.cn (G.J.); liaoyun@imbcams.com.cn (Y.L.); fst@imbcams.com.cn (S.F.); 13691256967@163.com (Z.N.); lidandan@imbcams.com.cn (D.L.); yuli@imbcams.com.cn (L.Y.); zhaoheng@imbcams.com.cn (H.Z.); xinglixu@imbcams.com.cn (X.X.); wlc@imbcams.com.cn (L.W.); cherryzhang629@126.com (Y.Z.)

**Keywords:** SARS-CoV-2, COVID-19, DNA vaccine, inactivated vaccine, variants, RBD

## Abstract

Objective: We constructed two DNA vaccines containing the receptor-binding domain (RBD) genes of multiple SARS-CoV-2 variants and used them in combination with inactivated vaccines in a variety of different protocols to explore potential novel immunization strategies against SARS-CoV-2 variants. Methods: Two DNA vaccine candidates with different signal peptides (namely, secreted and membrane signal peptides) and RBD protein genes of different SARS-CoV-2 strains (Wuhan-Hu-1, B.1.351, B.1.617.2, C.37) were used. Four different combinations of DNA and inactivated vaccines were tested, namely, Group A: three doses of DNA vaccine; B: three doses of DNA vaccine and one dose of inactivated vaccine; C: two doses of inactivated vaccine and one dose of DNA vaccine; and D: coadministration of DNA and inactivated vaccines in two doses. Subgroups were grouped according to the signal peptide used (subgroup 1 contained secreted signal peptides, and subgroup 2 contained membrane signal peptides). The in vitro expression of the DNA vaccines, the humoral and cellular immunity responses of the immunized mice, the immune cell population changes in local lymph nodes, and proinflammatory cytokine levels in serum samples were evaluated. Results: The antibody responses and cellular immunity in Group A were weak for all SARS-CoV-2 strains; for Group B, there was a great enhancement of neutralizing antibody (Nab) titers against the B.1.617.2 variant strain. Group C showed a significant increase in antibody responses (NAb titers against the Wuhan-Hu-1 strain were 768 and 1154 for Group C1 and Group C2, respectively, versus 576) and cellular immune responses, especially for variant B.1.617.2 (3240 (*p* < 0.001) and 2430 (*p* < 0.05) for Group C1 and Group C2, versus 450); Group D showed an improvement in immunogenicity. Group C induced higher levels of multiple cytokines. Conclusion: The DNA vaccine candidates we constructed, administered as boosters, could enhance the humoral and cellular immune responses of inactivated vaccines against COVID-19, especially for B.1.617.2.

## 1. Introduction

Coronavirus disease 2019 (COVID-19), caused by SARS-CoV-2, emerged in Wuhan, Hubei Province, China, at the end of 2019 [1] and was defined as a pandemic infectious disease by the World Health Organization (WHO) within only approximately 4 months [2]. As of 21 April 2022, SARS-CoV-2 has caused more than 500 million confirmed cases of COVID-19 and 6 million confirmed deaths [3]. SARS-CoV-2 is a novel single-stranded enveloped β-coronavirus RNA virus, encoding at least 29 proteins [4,5]. In the process of replication, SARS-CoV-2 has the characteristics of high mutagenicity of RNA virus, and has a great ability to spontaneously mutate. Since the virus was discovered, thousands of mutations have occurred [6], with 70 variants produced in approximately 24 months. These variants can be categorized roughly into five major lineages, namely, Alpha (9 variants), Beta (10 variants), Gamma (12 variants), Delta (8 variants), and Omicron (31 variants), with the first four (Alpha, Beta, Gamma, and Delta) proven to be more virulent and infectious than the original Wuhan strain (Wuhan-Hu-1) [7].

Vaccines designed for SARS-CoV-2 are known to be effective against COVID-19 by targeting the surface glycoprotein (S protein) or the receptor-binding domain (RBD) protein located within the S protein [8,9,10,11,12]. The full-length S protein is encoded by a 3822 bp ssRNA, while the RBD is only 669 bp in length. The small size of the RBD protein means not only that targeting this protein could lead to weak immunogenicity but also that the strategy of targeting the specific RBD sequences of various SARS-CoV-2 mutant strains was extremely feasible. Fortunately, the DNA vaccine route could make good use of this property of SARS-CoV-2. DNA vaccines utilize DNA plasmids as vectors to carry target genes encoding antigens into host cells (especially antigen presenting cells). This mechanism is based on the entry of genetic material into the nucleus. The mammalian promoter in the vector is then activated, and the transcription and translation of the target gene is performed by the host cells. This procedure not only has the capacity to activate humoral immunity but also can induce effective cellular immune responses [13,14]. In the COVID-19 pandemic, the DNA vaccines INO-4800 [15] and ZyCoV-D [16] (approved in India) showed potent activity against SARS-CoV-2 in animal experiments.

Inactivated vaccines, which contain whole or partially pathogenic viruses whose genetic materials have been destroyed, are a classic type of antiviral vaccine. As such, they are considered one of the safer vaccine options. They typically contain many proteins that the immune system can respond to, but because they cannot infect cells, inactivated vaccines can only stimulate antibody-mediated responses, which might be weaker and short-lived [17]. In a study of a SARS-CoV-2 inactivated vaccine, antiviral IgG titers declined at 10 weeks after vaccination [18]. To overcome this barrier, the inactivated vaccine is usually used with an adjuvant or boosted with a second dose [19,20].

When designing a vaccine, we should focus on producing an effective immune response, both humoral and cellular. In short, a well-designed vaccine must be able to activate not only humoral immune responses but also cellular immune responses. In the context of the continuous emergence of SARS-CoV-2 variants and the declining effectiveness of existing vaccines against novel variants [21,22,23,24,25], we built two DNA vaccine candidates with different signal peptides (namely, secreted signal peptides and membrane signal peptides) and target genes consisting of the RBD protein genes of four different SARS-CoV-2 strains: Wuhan-Hu-1, South Africa (B.1.351, Beta), Delta (B.1.617.2), Lambda (C.37) and Delta (B.1.617.2) (PS: there are two RBD genes of Delta strain). Then, the vaccines were administered in combination with SARS-CoV-2 inactivated vaccine under a variety of schedules (3 doses of DNA vaccine alone, 3 doses of DNA vaccine and 1 booster dose of inactivated vaccine, 2 doses of inactivated vaccine and 1 booster dose of DNA vaccine, or coadministration of 2 doses of DNA vaccine and inactivated vaccine) to identify a vaccination scheme incorporating the advantages of both DNA vaccines and inactivated vaccines and to explore their functional interactions.

## 2. Materials and Methods

### 2.1. Cell Lines and DNA Transfection

HEK-293T (ATCC^®^ CRL-3216^TM^) and African Green monkey kidney (Vero, ATCC, Old Town Manassas, VA, USA) cells were obtained from ATCC (Manassas, VA, USA). HEK-293T and Vero cell lines were cultured in DMEM (Corning, NY, USA) supplemented with 10% fetal bovine serum (Gibco, Grand Island, NY, USA) and penicillin–streptomycin (Gibco, USA). ExpiCHO cell lines were utilized for recombinant protein expression according to the manufacturer’s protocol (ExpiCHO™ Expression System Kit, Thermo, Waltham, MA, USA). DNA transfection into HEK-293T cells in vitro utilized jetPRIME^®^ (Polyplus-transfection, Illkirch-Graffenstaden, France).

### 2.2. Construction and Preparation of Recombinant Plasmid DNA

For our DNA vaccine candidates, the RBD protein gene from the Wuhan-Hu-1 strain (virus isolated from the respiratory secretions of an adult male patient at Yunnan Hospital of Infectious Disease in Kunming in January 2020) was obtained by PCR, and the RBD protein genes of three other variants (South Africa (B.1.351, Beta), Delta (B.1.617.2), and Lambda (C.37)) were synthesized by Sangon Biotech (Shanghai, China). They were covalently connected into pVAX-1 plasmid by endonuclease digestion. The RBD protein gene sequences, preceded by secreted or membrane signal sequences, were inserted into the pVAX-1 DNA vaccine vector (Invitrogen^TM^, Carlsbad, CA, USA). Then, the recombinant plasmid DNA was transformed into *E. coli* DH5α competent cells (TAKARA, Tokyo, Japan), harvested, and verified by first-generation sequencing techniques. WSDLD-T included the secreted signal peptide of the tPA (tissue-type plasminogen activator) protein and RBD proteins of the Wuhan-Hu-1, B.1.351, B.1.617.2, C.37 and B.1.617.2 strains. WSDLD-S differed from WSDLD-T in that the membrane signal peptide of the S protein was used as the signal peptide (the construction methods of the DNA vaccines are shown in Figure 1A). WSDL-T-EGFP and WSDL-S-EGFP included four RBD protein genes and one enhanced green fluorescent protein (EGFP) gene to indirectly verify that the gene of interest could be expressed in HEK-293T cells, and p-WSDL-T and p-WSDL-S recombinant plasmids (with a His-tag added to the C-terminus) were constructed with the same methods and the pVAX1 plasmid were replace with pcDNA3.1(+) to verify the target protein expression in suspension culture of ExpiCHO cells.

### 2.3. In Vitro Expression Analysis of the DNA Vaccine Candidates

Because a low level of proteins is produced by DNA vaccine-transfected cells in vitro, western blotting might not always directly detect its expression. Thus, we considered a variety of indirect methods to demonstrate its expression capacity. The green fluorescence of EGFP in HEK-293T cells and the SDS–PAGE and western blot analysis of the target proteins in ExpiCHO cells were utilized to show that the target genes could be expressed. HEK-293T cells were cultured in 6-well plates and transfected with 2 μg recombinant plasmids (WSDL-T-EGFP and WSDL-S-EGFP)/well using jetPRIME^®^ transfection reagent (Polyplus, Illkirch-Graffenstaden, France) following the manufacturer’s protocol. After 48 h, the expression of EGFP was observed by fluorescence microscopy. The target proteins expressed from the recombinant plasmids p-WSDL-T and p-WSDL-S were detected by SDS-PAGE and western blot assay.

### 2.4. Animals and Vaccination Programs

Six-week-old female BALB/c mice were purchased from Charles River (Beijing, China). Mice were randomly allocated into groups with 4 mice/group. Group A was constructed to evaluate the immune responses to DNA vaccine candidates alone, with A1 receiving WSDLD-T and A2 receiving WSDLD-S. Group A was immunized 3 times at 14-day intervals with a dose of 100 μg. Group B (B1, WSDLD-T; B2, WSDLD-S) received a booster dose of inactivated virus vaccine (30 U, 100 μL) in addition to the immunizations given to Group A (all vaccines were administered in intradermal injections in the back near the tail. In the immunization program of this research, all the DNA vaccines were 100 μg in 100 μL and inactivated vaccine was 30 U in 100 μL). Group C was immunized with two doses of inactivated DNA and boosted with a dose DNA vaccine 2 weeks later. Group D was immunized via the co-administration of inactivated vaccine (30 U) and DNA vaccine (100 μg), a total of 200 μL/dose. The subgroups of Group C and Group D were the same as those of Groups A and B. Serum samples were collected at day 0 and 14 after the last immunization. All vaccination programs are shown in Figure 1B.

### 2.5. ELISA

The S1 protein (Sanyou Biopharmaceuticals Co., Ltd., Shanghai, China) of the SARS-CoV-2 Wuhan-Hu-1 strain was utilized to coat 96-well ELISA plates (Corning, NY, USA) at a concentration of 0.1 μg/well and incubated at 4 °C overnight. Then, the plates were blocked with 1% BSA–phosphate-buffered saline (PBS) and visualized with horseradish peroxidase-conjugated goat anti-mouse IgG (Thermo Fisher, USA) and TMB (3,3′,5,5′-tetramethylbenzidine) substrate (Solarbio, Beijing, China) according to previously described methods [26]. The reaction was evaluated at 450 nm by an ELISA plate reader (Gene Company, Hong Kong, China), and the S1-specific IgG titers were determined by end titration utilizing the reciprocal of the lowest serum dilution that produced an OD value 2.1-fold greater than that in the prebleed.

### 2.6. ELISPOT Assay

The spleens were collected from immunized mice and processed into single-cell suspensions in RPMI 1640 medium (Gibco, USA). An ELISPOT assay was performed with the mouse IFN-γ ELISPOTPLUS kit (ALP) (Mabtech, Nacka Strand, Sweden) and mouse IL-4 ELISPOTPLUS kit (ALP) (Mabtech, Sweden) in accordance with the manufacturer’s protocols. The pre- and postvaccination splenocytes were stimulated in vitro by RBM and non-RBM peptide pools spanning the RBD region of the Wuhan-Hu-1 strain, composed of 15-mer peptides with 11 amino acids (aa) of overlap (Sino Biological, Beijing, China). For each sample, four wells were prepared: one positive control (cells + phytohemagglutinin (PHA)), two sample wells (cells + 15-mer peptides), and one negative control (cells + serum-free medium universal (DAKAWE, Wuhan, China)). The working volume of each well was 200 μL, and there were 4 blank control wells (200 μL serum-free media only) in every plate (96 wells). The plate was incubated at 37 °C with 5% CO_2_ for 36 h, after which time the cells were removed, the plates were washed, and the spots were developed according to the manufacturer’s protocol. An ELISPOT reader (CTL, Shaker Heights, OH, USA) was utilized to count the colored spots. The calculation of spot-forming units (SFUs) per million cells entailed subtracting the negative control wells.

### 2.7. Neutralization Antibody Assays

Two kinds of neutralization assays were tested owing to that we did not have the real virus of all the SARS-CoV-2 variants in the paper. First, the neutralizing activities against native SARS-CoV-2 virus (Wuhan-Hu-1 strain) were conducted at IMB (Institute of Medical Biology, Chinese Academy of Medicine Sciences, Beijing, China) according to previously described methods [27]. Replication-deficient recombinant SARS-CoV-2 pseudoviruses (rVSV-SARS-CoV-2) produced from VSV (vesicular stomatitis virus) and bearing the B.1.351, B.1.617.2 and C.37 variants were purchased from Vazyme Biotech Co., Ltd., Nanjing, China. In the pseudovirus system, the SARS-CoV-2 spike protein expressed on the surface of rVSV forms chimeric virus particles, which can bind to endogenous ACE2 (angiotensin converting enzyme 2)-expressing cells (Vero), closely simulating the process of SARS-CoV-2 invasion of target cells through spike protein binding to ACE2. The pseudovirus also expresses luciferase, an enzyme that is easy to detect. To determine the neutralization activity of the serum sample, threefold serial dilutions were performed for heat-inactivated serum samples in duplicate with the starting range from 1:20 to 43,740, adding 650 TCID_50_ (tissue culture infective dose 50%) pseudovirus per well in accordance with the manufacturer’s protocol.

### 2.8. Flow Cytometry

Lymphocytes from local lymph nodes were collected and stained with different fluorescence-labeled antibody cocktails as we previously described [27] to detect the percentages of CD4^+^ T cells (CD3-FITC, CD4-PE), CD8^+^ T cells (CD3-FITC, CD8a-APC), IFN-γ secreting cells (CD3-FITC, CD8-APC, IFN-γ-PE-Cy5-5), Tfh (T follicular helper) cells (CD45-APC-Cy7 (BD Pharmingen, San Diego, CA, USA), CD4-FITC (BD Pharmingen, USA), CD185-APC, PD-1-Pacific Blue (BD Pharmingen, USA)), germinal center (GC) B cells (B220-APC-Cy7, CD45-APC-Cy7 (BD Pharmingen, USA), CD95-PE, GL-7-APC), and plasma cells (B220-Pacific Blue, CD27-PE-Cy7, CD138-PE). Except where indicated, the remaining antibodies were purchased from BioLegend, San Diego, CA, USA. The stained cells were assessed by flow cytometry (BD LSRFortessa ^TM^ Cell Analyzer, San Jose, CA, USA), followed by FlowJo software analysis.

### 2.9. Proinflammatory Response

The proinflammatory cytokine levels in serum samples were analyzed. Mouse IL-1β, TNF-α, IFN-γ, IL-2, IL-4, IL-5, IL-6, IL-10, CXCL1 (C-X-C motif ligand 1) and IL-12p70 (Interleukin 12p70) were measured by a Bio-Plex Luminex xMAP-based multiplex bead-based immunoassay (Bio–Rad, San Francisco, CA, USA). During detection, the serum sample and the reporter molecule were added successively to react with the labeled microspheres. The target molecules (antigen to be detected) in the serum specifically combine with the probe and reporter molecule such that the microspheres of the cross-linked probe become carriers of the reporter molecule, phycoerythrin. Then, the microspheres were detected and analyzed by Bio-Plex Manager™ software.

### 2.10. Statistics

All statistical analyses were conducted in one-way ANOVA (and nonparametric or mixed) using GraphPad Prism 8 (GraphPad software). Data were defined as statistically significant if the *p* value was <0.05.

## 3. Results

### 3.1. Expression of SARS-CoV-2 DNA Vaccine Candidates

At 48 h post-transfection, transfected HEK-293T cells were observed under a fluorescence microscope to analyze the green fluorescence of EGFP. As expected, EGFP was expressed after transfection with WSDL-T-EGFP and WSDL-S-EGFP (Figure 2A,B). Furthermore, the expression of the target proteins from p-WSDL-T and p-WSDL-S recombinant plasmids, as tested by SDS–PAGE and western blot assays, also proved that the DNA vaccine candidates could be expressed using the construction methods in the paper (Figure 2C,D).

### 3.2. Humoral Immune Responses

Intradermal immunization with two DNA vaccine candidates administered alone (Group A1 and Group A2) elicited weak serum IgG responses against the S1 protein and few neutralizing antibodies (NAbs) against the wild-type strain (Wuhan-Hu-1) and pseudovirus variants (Figure 3A). However, when an additional dose of inactivated vaccine (30 U, intradermal route) was added as a booster dose (Group B1 and Group B2), the antibody responses showed a greater increase than in Group A and in controls receiving one dose of inactivated vaccine (Figure 3A). The extremely rapid rate of mutation in the SARS-CoV-2 virus reduced the effectiveness of inactivated vaccines developed using earlier strains (Wuhan-Hu-1). The combination of vaccination with the SARS-CoV-2 inactivated vaccine and DNA vaccines containing the RBD protein gene sequences of multiple SARS-CoV-2 variants significantly increased the protection potential against the wild-type SARS-CoV-2 strain as well as several variants. For Group C (C1 and C2), the IgG titers of both subgroups exhibited an approximately 12-fold increase compared to the inactivated vaccine (two doses) (*p* < 0.05, 0.033 and 0.028, respectively), with a smaller increase in neutralizing antibodies against the Wuhan-Hu-1 strain (Figure 3B). For Group D (D1 and D2), the antibody expression level increase compared to 2 doses of inactivated vaccine was not significant (Figure 3A,B). For the SARS-CoV-2 variant pseudovirus neutralization assay, the level of neutralizing antibodies targeting the B.1.351 and C.37 strains increased in Group C and Group D, compared to two doses inactivated vaccine group. Moreover, as shown in Figure 3D, the NAb titers indicated that the two DNA vaccine candidates had potent activity against the Delta (B.1.617.2) strain.

### 3.3. Cellular Immune Responses

Splenocytes of immunized mice collected at 2 weeks after the last immunization utilizing the pooled RBD peptides of the SARS-CoV-2 of Wuhan-Hu-1 strain as a stimulus were evaluated to detect IFN-γ and IL-4 secreting cells. Mice that were immunized with DNA vaccine only (Group A) or immunized with inactivated virus as a booster (Group B) did not produce IFN-γ- and IL-4-secreting positive cells after stimulation with the pooled peptide. However, the protocol for Group C (C1 and C2) elicited stronger cellular immune responses than the 2 inactivated doses (*p* < 0.05, 0.003 and 0.0001 for IFN-γ-secreting cells and 0.0069 and 0.0137 for IL-4-secreting cells, respectively). Compared to the 2-dose inactivated vaccine, the protocol for Group D also yielded a certain increase in cellular immunity (Figure 4A,B).

### 3.4. Detection of Immune Cell Populations and Cytokine Expression

In vaccine research and development, it is required not only to meet the criterion of high immunogenicity but also to evaluate the mechanism of the vaccine. Therefore, we sought to understand the responses of various immune cells and proinflammatory cytokines after the delivery DNA vaccine candidates and the interactions between the vaccines and the immune system. As shown in Figure 5, there were no significant differences between groups in terms of the percentage of CD4+ T and CD8+ T cells. Remarkably, the treatments of Group B (B1 and B2) produced higher percentages of Tfh cells and GC B cells. The percentages of IFN-γ-secreting cells and plasma cells were highest in Group C (Figure 5). The cytokine levels produced by different vaccine groups are shown in Figure 6. Serum cytokine levels were measured when the mice were sacrificed. Overall, Group C exhibited a higher level of cytokine immune response (i.e., higher levels of IL-1β, TNF-α, IFN-γ, IL-2, IL-4, IL-6, IL-10 and IL-12p70), which partly explains why Group C generated stronger immune responses than the other groups. Although the two DNA vaccine candidates consisting of different signal peptides (secreted and membrane signal peptides) showed no difference in humoral or cellular immune responses, the DNA vaccines containing secreted signal peptides were found to induce higher levels of IL-1β, TNF-α, IL-2, IL-4, IL-5, IL-6, IL-10 and IL-12p70, whereas the membrane signal peptides induced higher expression levels of the cytokine CXCL1.

## 4. Discussion

As an emerging infectious virus, SARS-CoV-2 has caused an enormous social panic and public health crisis across the globe. Vaccination has been considered a key strategy to control COVID-19 [28,29]. In the context of this SARS-CoV-2 pandemic, vaccine scientists have seen the fastest ever development of a vaccine (namely, an mRNA vaccine), from laboratory design to marketing approval within one year, the reasons for its rapid success might owing to mRNA vaccine technology route bypass many of the factors that need a lot of time to develop traditional classic vaccines, such as virus seed screening, cell culture and high-standard manufacturing facilities. However, the mRNA vaccine is manufactured from DNA plasmids, whereas a DNA vaccine might be manufactured more quickly and therefore much more responsive to an emerging infectious disease. In the midst of the COVID-19 pandemic, nucleic acid vaccines, including mRNA vaccines and DNA vaccines, were successfully approved for marketing for the first time, indicating the feasibility of DNA vaccines.

Since the emergence of the wild-type SARS-CoV-2 strain Wuhan-Hu-1, several new variants of concern have developed, such as B.1.351 (Beta, first reported in South Africa in December 2020), B.1.617.2 (Delta, first reported in India in December 2020), and C.37 (Lambda, first detected in Peru in February 2021) [30]. The vast majority of the current COVID-19 vaccines were designed based on the original Wuhan-Hu-1 strain. In the early stages of the pandemic, these vaccines, including inactivated vaccines [31,32,33], mRNA vaccines [10,34,35], viral vector vaccines [9,36,37], protein subunit vaccines [11,38,39], and DNA vaccines [40,41], all provided satisfactory protection. In a meta-analysis comparing the effectiveness of different COVID-19 vaccines, the pooled effectiveness of inactivated vaccine was only 61% (95% CI: 52–68%), which was relatively low [42]. Other studies have also shown that inactivated vaccines are less effective than mRNA vaccines, viral vector vaccines and protein subunit vaccines; this phenomenon is related to the unique mechanism of action of inactivated vaccines [42,43,44], namely, weaker antibody responses, short life, and weak ability to induce cellular immunity. Additionally, there have been two more difficult problems to address. One is that antibody titers elicited by COVID-19 vaccines declined quickly, and the ability to target certain SARS-CoV-2 variants decreased even more quickly [45,46,47]. The other is that new variants are constantly emerging and becoming the main infectious strains in different regions [48,49,50]. Thus, considering that inactivated vaccines are currently among the most widely used vaccines in the world, and considering their inherent disadvantages, we constructed two DNA vaccine candidates that could cover more variants of SARS-CoV-2 and tested their administration with inactivated vaccines in various combinations. This approach allows us to realize the advantages of DNA vaccines, which can be continuously expressed in the body and induce strong cellular immune responses, to supplement the shortcomings of inactivated vaccines and achieve a better vaccination strategy for COVID-19 [13,14,51]. The inactivated vaccine that we used has been approved for emergency use in China [27,52].

Because low levels of protein are produced by DNA vaccine-transfected cells in vitro, western blotting may not always directly detect its expression. Therefore, in our research, we confirmed the expression of the DNA vaccines through two indirect methods: one was to add the EGFP gene sequence following the target gene sequence and then confirm the expression by observing the green fluorescence; the other was to produce more target proteins in cells in suspension cell in vitro to allow their detection by SDS–PAGE and western blot assays. The immunogenicity of two DNA vaccine candidates (WSDLD-T and WSDLD-S) administered alone (Group A1 and Group A2) via intradermal delivery did not produce satisfactory results. There were no significant differences between the subgroups using different signaling peptides (secreted protein signal and membrane protein signal). While utilizing the DNA vaccine as a booster for two doses of inactivated vaccine, the cellular immune response showed greatly enhanced; and the antibody responses (IgG titer and NAb titer) against the Wuhan-Hu-1 strain and B.1.617.2 showed large and statistically significant improvements. However, for B.1.351 and C.37, there was no significant difference in the increase in NAb titer. Interestingly, Group B, using an inactivated vaccine as a booster for 3 doses of two DNA vaccine candidates, elicited a higher percentage of Tfh cells and GC B cells than other groups. Considering that Tfh and GC B cells are located in the GC, which is the site for the production of high-affinity antibodies, we speculated that utilizing an inactivated vaccine as a booster for DNA vaccines might be effective in improving antibody affinity. In addition, we found that DNA vaccine candidates containing secreted signal peptides induced higher levels of expression of multiple cytokines in this immunization strategy, suggesting that different signal peptides should be considered in vaccine design. It is worth noting that inflammatory cytokine levels are dynamic in real time, and the serum samples we tested were limited to those collected after sacrificing mice after completing the immunization procedure, which therefore could provide only a relatively limited reference value. Notably, owing to the specific characteristics of the mouse model, we were able to indirectly explore the safety of the DNA vaccine candidates in this study by measuring the body weight of the treated mice (Appendix A). No deaths occurred in mice any DNA vaccine group.

However, there were some limitations in the paper. Firstly, we only tested in small animal models of mice, not in large animal models such as monkeys. And then, owing to the right conditions, we did not conduct the real virus neutralization assay and cross protection tests on all the strains. The other limitation was that we did not fully explore the safety of DNA vaccines in mice.

The rapid development of a safe and effective COVID-19 vaccine was a remarkable achievement for mankind. However, continuously emerging SARS-CoV-2 variants and the possibility of decreased immunity following vaccination have prompted the need for additional immunization strategies [53]. This has put further pressure on those in neglected low-income countries, where access to COVID-19 vaccines was already inadequate. For booster vaccine doses, evidence has shown that a heterologous prime-boost strategy was more effective than a homologous prime-boost approach [54]. The advantage of heterologous prime-boost enhanced immunity is that each delivery system induces humoral and cellular immunity against specific antigens. For example, inactivated vaccines and subunit vaccines mainly cause humoral immune responses, while DNA vaccines and recombinant live vectors effectively induce cell-mediated immunity [55,56,57]. Researchers explored the effectiveness of heterologous (mRNA vaccine or recombinant adenoviral vector vaccine) versus homologous (inactivated vaccine) COVID-19 booster vaccination in volunteers who previously received two doses of COVID-19 inactivated vaccines. They found that while the antibody titers were low at six months after the previous two doses of inactivated vaccine, the booster dose induced a significant increase in binding and neutralizing antibodies, which might improve the protection against infection. More importantly, a heterologous booster dose resulted in stronger immune responses than a homologous booster and might enhance the protective effect [58] and produce higher antibody levels of IgM and IgA [59]. Additionally, even in some specific populations, such as kidney transplant recipients who had received two doses of mRNA vaccine but did not develop antibodies, a booster dose vaccination increased the likelihood of developing antibodies [60].

In summary, SARS-CoV-2 inactivated vaccines have been among the most widely used vaccines in the COVID-19 pandemic, especially in low- and middle-income countries [61]. COVID-19 DNA vaccines are easy to scale, inexpensive, and can include genetic information from various variants. For mice that received a complete course (two doses) of inactivated vaccine, the DNA vaccine candidates that we constructed could improve the IgG antibody responses and neutralizing antibody titers. Utilizing heterologous DNA vaccines as a booster showed greater improvement than coadministration, especially against the Delta strain (B.1.617.2), and these two DNA vaccines all could greatly improve the cellular immune response, which indicated that this approach has potential for constructing vaccines against SARS-CoV-2 variants. The low immune responses produced by DNA vaccine candidates administered alone indicates that further research will be needed to increase their expression capacity in vivo.

## Figures and Tables

**Figure 1 vaccines-10-00929-f001:**
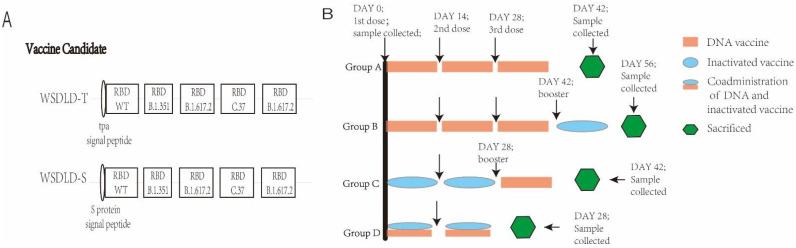
(**A**) Schematic diagram of two DNA vaccine candidate constructs. RBD: receptor binding domain. (**B**) Animal immunization and sample collection procedures. The DNA vaccines all contained two RBD genes of B.1.617.2 (Delta strain).

**Figure 2 vaccines-10-00929-f002:**
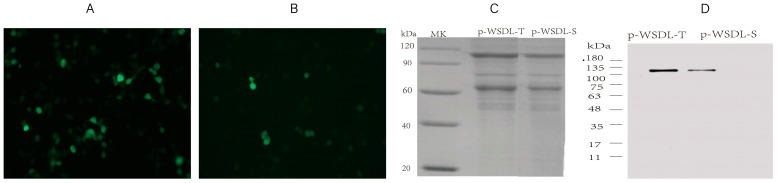
Protein expression analysis: (**A**) The expression of recombinant plasmid WSDL-T-EGFP in transfected HEK293T cells. (**B**) The expression of recombinant plasmid WSDL-S-EGFP in transfected HEK293T cells. Images were analyzed with a fluorescence microscope with a 20× objective. (**C**) SDS–PAGE and (**D**) western blot assay to detect the purified protein of DNA vaccines (p-WSDL-T and p-WSDL-S) bearing four RBD regions expressed in ExpiCHO cell lines.

**Figure 3 vaccines-10-00929-f003:**
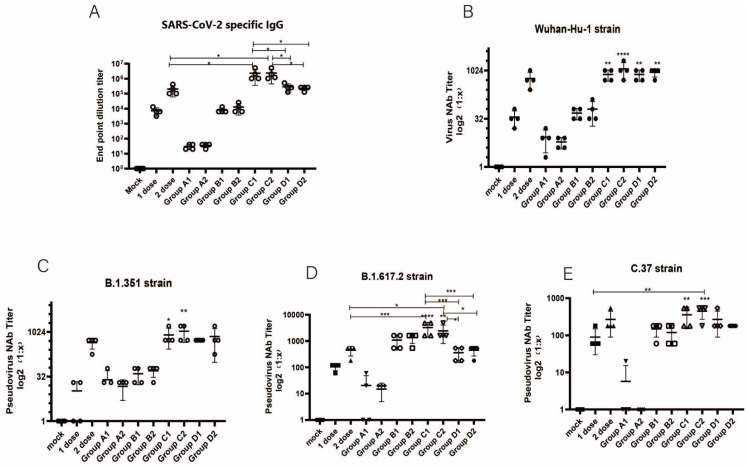
Antibody responses to different group designs: (**A**) Titers of SARS-SoV-2 S1-specific total IgG analysis (Wuhan-Hu-1 strain). (**B**–**E**) Neutralizing antibody (NAb) titers against the (**B**) Wuhan-Hu-1, (**C**) B.1.351, (**D**) B.1.617.2, and (**E**) C.37 strains. The latter three variants were tested by pseudovirus neutralization assays. 1 dose: 1 dose inactivated vaccine control group; 2 dose: 2 doses inactivated vaccine control group; one-way ANOVA (and nonparametric or mixed) was conducted; * *p* < 0.05; ** *p* < 0.01; *** *p* < 0.001; **** *p* < 0.0001 versus the control group.

**Figure 4 vaccines-10-00929-f004:**
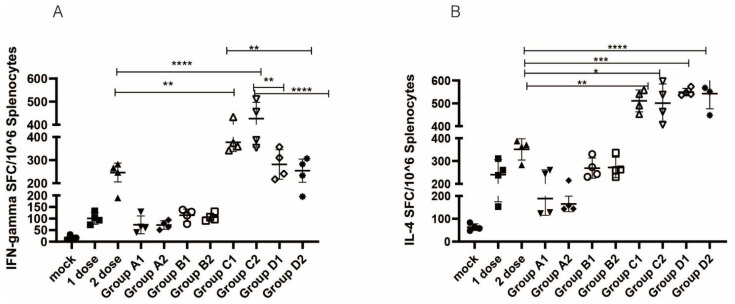
SARS-CoV-2 RBD-specific T-cell responses analyzed by ELISPOT: (**A**) RBD-specific IFN-γ and (**B**) RBD-specific IL-4 responses were measured. 1 dose: 1 dose inactivated vaccine control group; 2 dose: 2 doses inactivated vaccine control group; one-way ANOVA (and nonparametric or mixed) was conducted; * *p* < 0.05; ** *p* < 0.01; *** *p* < 0.001; **** *p* < 0.0001 versus the control group.

**Figure 5 vaccines-10-00929-f005:**
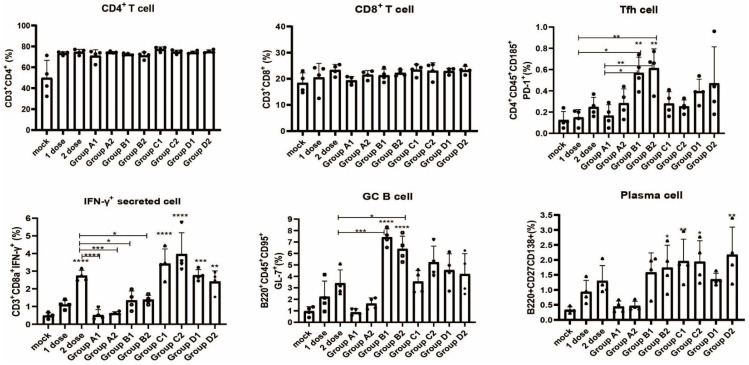
Different vaccination groups elicited immune cell population percentage changes in CD4+ T cells, CD8+ T cells, Tfh cells, GC B cells, plasma cells and IFN-γ-secreting cells in local lymph nodes. For cell data detected as 0% in flow cytometry, we assigned a value of 0.05% for plotting. 1 dose: 1 dose inactivated vaccine control group; 2 dose: 2 doses inactivated vaccine control group; one-way ANOVA (and nonparametric or mixed) was conducted; * *p* < 0.05; ** *p* < 0.01; *** *p* < 0.001; **** *p* < 0.0001 versus the control group.

**Figure 6 vaccines-10-00929-f006:**
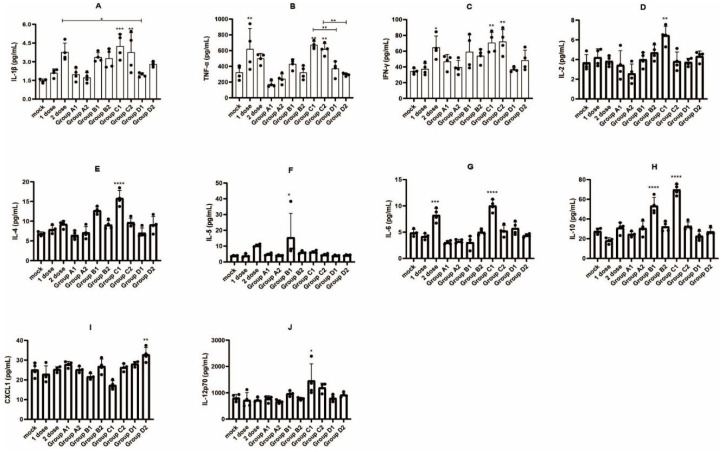
Proinflammatory cytokine analysis of serum samples collected from mice at the time of sacrificing after completion of the immunization procedure. (**A**) IL-1β, (**B**) TNF-α, (**C**) IFN-γ, (**D**) IL-2, (**E**) IL-4, (**F**) IL-5, (**G**) IL-6, (**H**) IL-10, (**I**) CXCL1, and (**J**) IL-12p70. 1 dose: 1 dose inactivated vaccine control group; 2 dose: 2 doses inactivated vaccine control group; one-way ANOVA (and nonparametric or mixed) was conducted; * *p* < 0.05; ** *p* < 0.01; *** *p* < 0.001; **** *p* < 0.0001 versus the control group.

## Data Availability

Not applicable.

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
