# Peer review of "Immunological Study of Combined Administration of SARS-CoV-2 DNA Vaccine and Inactivated Vaccine"

_vaccines, 2022, doi:10.3390/vaccines10060929_

Round 1

Reviewer 1 Report

Thank you for considering me to review the manuscript:

Immunological study of combined administration of SARS- 2 CoV-2 DNA vaccine and inactivated vaccine

The global pandemic of COVID-19 has presented a significant threat to public health worldwide. Vaccination has been considered a key strategy to control COVID-19. However, the emergence of coronavirus variants is likely to impact vaccine efficacy. Therefore there is a need to develop new strategies for vaccination. This study shows that using DNA vaccine as a booster could enhance the inactivated vaccine's humoral and cellular immune responses. In this manuscript, the authors constructed two DNA vaccines containing RBD genes of multiple SARS-CoV-2 variants. They used them as a booster with inactivated vaccines to explore potential immunisation strategies against SARS-CoV-2 variants.

Overall, I found the manuscript to be of good quality and relevant to the field. The experimental design is appropriate; however, details are required to be reproducible.

There are a few points I would recommend to the authors for consideration:

Abstract

• Line 10: (RBD) Abbreviation is not defined

• Line 24: (NAb) Abbreviation is not defined

Introduction

• Line 68: what do the authors mean by imbalance of different immune cells. Consider referring to a different reference.

• Line 77 & 78: Wuhan-Hu-1, 76 South Africa (B.1.351, Beta), Delta (B.1.617.2), Lambda (C.37) and Delta (B.1.617.2).

Why are you repeating Delta (B.1.617.2) in the sentence?

• Line 77-81: Then, the vaccines were administered in combination with SARS-CoV-2 inactivated vaccine under a variety of schedules (3 doses of DNA vaccine and 1 booster dose of inactivated vaccine, 2 doses of inactivated vaccine and 1 booster dose of DNA vaccine, or coadministration of 2 doses of DNA vaccine and inactivated vaccine).

There is no mention of the three doses of DNA vaccine on their own.

Material & methods

• Line 102: (WSDLD-T) Abbreviation is not defined.

• Line 104: the Wuhan-Hu-1, B.1.351, B.1.617.2, C.37 and B.1.617.2 strains.

Why are you repeating B.1.617.2 strains in the sentence? Also, in fig 1A.

• Line 104: (WSDLD-S) Abbreviation is not defined.

• Line 93: 2.2. Construction and preparation of recombinant plasmid DNA

There is no reference or description of the method used to construct recombinant plasmid.

• Line 121: Polyplus; the country of the manufacturer is required.

• Line 136 & 137: (all vaccines were administered in intradermal injections in the back 136 near the tail, with DNA vaccine 100 μg in 100 μL, inactivated vaccine in 100 μL (30 U)).

Inactivated vaccine needs to be removed because Group A is only a DNA vaccine.

• Figure 1B: does not show the type of doses given to the mice, whether they are DNA or inactivated vaccines. In addition, no mention that serum samples were collected on day 0.

• Line 176: the authors did not explain why they tested two neutralisation assays.

• Line 180: (VSV) Abbreviation is not defined.

• Line 181: Vazyme Biotech Co., Ltd; the country of the manufacturer is required.

• Line 183: (ACE2) Abbreviation is not defined.

• Line 188: (TCID50) Abbreviation is not defined.

• Line 189: There is no reference or description of the method used for the flow cytometry.

• Line 193, 194 & 195: BD Pharmingen; the country of the manufacturer is required.

• Line 198: BD LSRFortessa TM Cell Analyzer; the country of the manufacturer is required.

• Line 202: (CXCL1 and IL-12p70) Abbreviation is not defined.

• Line 209: 3. Statistics suggested being changed to 2.10 Statistics as part of materials and methods.

Accordingly, the numbering of the following sections should be changed:

- 4. Results to 3. Results

- 5. Discussion to 4. Discission

There is no mention of the statistical methods utilised to analyse the data.

Results

• Figure 3: statistical analysis that is demonstrated by stars is not mentioned in the legend.

What does 1 dose and 2 dose means? Also, in fig 4, fig 5 & fig 6.

The figure legend should be self-explanatory. The same for fig 4, fig 5 & fig6.

• Line 246 & 247: strains increased in Group C and Group D, although the increase relative to that induced by the 2-dose inactivated vaccine was not statistically significant.

I am confused; if it is not significant, what do the stars mean?

• Figure 4: the A & B are not shown in the figure.

• Supplementary Materials: I couldn’t access it through the provided link (www.mdpi.com/xxx/s1)

Reviewer 2 Report

In this article, the authors deal about Immunological study of combined administration of SARS- 2 CoV-2 DNA vaccine and inactivated vaccine. The authors constructed two DNA vaccines containing the RBD genes of multiple 10 SARS-CoV-2 variants and used them in combination with inactivated vaccines in a variety of differ- 11 ent protocols to explore potential novel immunization strategies against SARS-CoV-2 variants.

Introduction: Lines 35-44 add more data about the genome of SARS-CoV-2 virus. Lines 45-83:

The authors describe the mechanisms of COVID-19 vaccines made so far compared to those designed by them. Because they are difficult to follow, I suggest a diagram with these clearly highlighted mechanisms, which should be presented comparatively.

Figure 1. It is made in gray/black scale, it does not have good visibility, it is difficult to understand, I suggest re-editing it in color and with larger, legible letters.

Lines 299-300: briefly mention the stages of obtaining these vaccines in one year compared to the classic vaccines obtained in 10-15 years.

It is not clear enough in the discussion that DNA-based vaccines are clearly less potent than mRNA vaccines and these need higher doses to stimulate an immune response against SARS-CoV-2. Revise it.

Mention the limitation of this study.

What perspectives for human health does this MS have?

Consider revision accordingly.

Reviewer 3 Report

The article has done an important aspect of research regarding Covid19 vaccine efficacy. The different strategies have been compared to show the best vaccine strategy. I have following comments for the author,

1. The author has mentioned that the Group A has "weak" cellular and antibody responses. What does he mean by weak? There is still detectable immune responses as shown in the figures.

2. Line 66-67 of the introduction section needs reference for the statement.

3. Why there is no group where only inactivated vaccine has been used? In my opinion, this is necessary since it can show how much fold protective antibody can be produced using just inactivated vaccine strategy.

4. It would be great if the author explains whether all the assays for different groups were performed at the same time?

5. Figure 1 (B): Group C sample was collected after 14 days of last vaccine i.e. 42 days. This needs to be corrected.

6. Figure 2 (A) & (B) should be clearer. It is very hard to recognize the cells in fluorescent picture.

7. The author has shown the immune responses post 14 days of the booster. But it is possible that the immune responses could be better in later days for other groups. The author needs to explain why 14 days immune responses have been checked. Samples can be collected after 14 days as well and it would great to characterize the immune responses post 14 days.

8. Memory B cells are also necessary for vaccine mediated immune protection. It would be great to include the status of memory B cells post booster dose.

9. Figure 4, has two graphs and they need to be identified as A and B.

10. Figure 4 (B), for IL-4 ELISpot, one mouse from Group D2 has been removed i.e. the analysis has been done with three mice instead of four. The author needs to explain why the mouse has been removed.

Round 2

Reviewer 1 Report

The manuscript has been sufficiently improved for publication.

Reviewer 2 Report

No answer given.

Reviewer 3 Report

The author has answered all the concerns and it can be published now. Also, the author has made the changes suggested.